# Evaluation of In Vitro Capsaicin Release and Antimicrobial Properties of Topical Pharmaceutical Formulation

**DOI:** 10.3390/biom11030432

**Published:** 2021-03-15

**Authors:** Enkelejda Goci, Entela Haloci, Antonio Di Stefano, Annalisa Chiavaroli, Paola Angelini, Ajkuna Miha, Ivana Cacciatore, Lisa Marinelli

**Affiliations:** 1Pharmacotherapeutic Research Center, Faculty of Medical Sciences, Aldent University, 1001 Tirana, Albania; miha.ajkuna05@gmail.com; 2Department of Pharmacy, Faculty of Medicine, University of Medicine, 1001 Tirana, Albania; entela.haloci@umed.edu.al; 3Department of Pharmacy, University of “G. d’Annunzio” Chieti-Pescara, 66100 Chieti, Italy; adistefano@unich.it (A.D.S.); annalisa.chiavaroli@unich.it (A.C.); ivana.cacciatore@unich.it (I.C.); l.marinelli@unich.it (L.M.); 4Department of Chemistry, Biology and Biotechnology, University of Perugia, 06100 Perugia, Italy; paola.angelini@unipg.it

**Keywords:** capsaicin, hydrogel, antimicrobial, in vitro release, antifungal

## Abstract

(1) Background: Capsaicin is the main capsaicinoid of the *Capsicum* genus and it is responsible for the pungent taste. Medical uses of the fruits of chili peppers date from the ancient time until nowadays. Most of all, they are used topically as analgesic in anti-inflammatory diseases as rheumatism, arthritis and in diabetic neuropathy. Reports state that the *Capsicum* genus, among other plant genera, is a good source of antimicrobial and antifungal compounds. The aim of this study was the preparation of a pharmaceutical Carbopol-based formulation containing capsaicin and the evaluation of its in vitro release and antimicrobial and antifungal properties. (2) Methods: It was first stabilized with an extraction method from the *Capsicum* annuum fruits with 98% ethanol and then the identification and determination of Capsaicin in this extract was realized by HPLC. (3) Results and Conclusions: Rheological analyses revealed that the selected formulation exhibited a pseudo-plastic behavior. In vitro release studies of capsaicin from a Carbopol-based formulation reported that approximately 50% of capsaicin was release within 52 h. Additionally, the Carbopol-based formulation significantly increased the antimicrobial effects of capsaicin towards all tested bacteria and fungi strains.

## 1. Introduction

Capsaicin, a pungent compound in chili peppers, is a highly selective agonist for the transient receptor potential vanilloid 1 receptor expressed in nociceptive sensory nerves [1]. It has been employed topically to treat various diseases such as rheumatoid arthritis, osteoarthritis, diabetic neuropathy and to relieve post-operative pain [2,3].

The two most abundant capsaicinoids in peppers are capsaicin (8-methyl-N-vanillyl-trans-6-nonenamide) and dihydrocapsaicin, both constituting about 90%, with capsaicin accounting for ~71% of the total capsaicinoids in most of the pungent varieties [4].

Given the limitations of current oral and parenteral therapies for the management of pain arising from various forms of nerve injury, alternate therapeutic approaches that are not associated with systemic adverse events that limit quality of life, impair function, or threaten respiratory depression are critically needed. Moreover, neuropathic conditions can be complicated by progressive changes in the central and peripheral nervous system, leading to persistent reorganization of pain pathways and chronic neuropathic pain. Recent advances in the use of high-dose topical capsaicin preparations hold promise in managing a wide range of painful conditions associated with peripheral neuropathies and may in fact help reduce suffering by reversing progressive changes in the nervous system associated with chronic neuropathic pain conditions [5].

Topical creams with capsaicin are used to treat pain from a wide range of chronic conditions including neuropathic pain. Following application to the skin capsaicin causes enhanced sensitivity to noxious stimuli, followed by a period with reduced sensitivity and, after repeated applications, persistent desensitization. There is uncertainty about the efficacy and tolerability of capsaicin for treating painful chronic neuropathies [6].

When applied locally to skin, it promotes an analgesic response due to the desensitizing of sensory neurons caused by substance P depletion [7].

Capsaicin is a lipophillic molecule with high degree of first pass metabolism and the short half-life when taken orally or by intravenous administration which made topical application of it a great advantageous for local and systemic effects. 

Beside its multiple pharmacological properties, capsaicin has recently also attracted attention for its antimicrobial activities [8]. Currently, researchers have shown that capsaicin can inhibit the growth of some food-borne pathogenic microorganisms, such as *Listeria monocytogenes*, *Helicobacter pylory*, *Pseudomonas aeruginosa*, *Botrytis cinerea*, *Aspergillus niger* and *Penicillium expansum* [9,10,11,12,13], whereas an anti-virulence activity has been demonstrated against *Vibrio cholerae*, *Staphylococcus aureus* and *Poephyromonas gingivalis* [14,15,16,17]. 

A study by Nascimento and colleagues (2014) tested the antimicrobial activity of capsaicin against Gram negative bacteria (*Escherichia coli*, *Pseudomonas aeruginosa*, and *Klebsiella pneumoniae*), Gram positive bacteria (*Enterococcus faecalis*, *Bacillus subtilis*, and *Staphylococcus aureus*) and one yeast (*Candida albicans*). The minimal inhibitory concentration (MIC) values obtained ranged from 0.06 to 25 μg/mL [18]. 

Marini et al. (2015) reported that the MICs for capsaicin to inhibit erythromycin-resistant and erythromycin-susceptible strains were prevalently in a narrow range (0.064–0.128 mg/mL), where the most common MIC was 0.128 mg mL^−1^ and the highest MIC was exhibited at 0.512 mg/mL [19]. Moreover, Ndashe et al. (2020) showed that the minimum bactericidal concentration (MBC) of capsaicin on the growth of *Lactococcus garvieae* was 0.1967 mg/mL, according with other studies [20,21]. 

Chromatographic methods, Thin Layer Chromatography (TLC)/High Performance Thin Layer Chromatography (HPTLC) and High Performance Liquid Chromatography (HPLC) TLC/HPTLC and HPLC are used extensively for qualitative and quantitative determination of this principle from the *Capsicum* herb and pharmaceutical products (capsaicin gels) [22,23,24].

Our study focuses on the formulation of topical capsaicin ethanolic extracts in pharmaceutical gels with prolonged effects avoiding its hepatic metabolism and improving bioavailability. Thus, we prepared a pharmaceutical gel based on Carbopol containing capsaicin and evaluated the in vitro release of the active principle. The quantitative determination of capsaicin from extracts and also from Carbopol-based gel formulations were performed by HPLC as described by Abdullah Al Othman Z. et al., with some slight modifications [25]. Antimicrobial effects against multiple bacterial and fungal strains were evaluated, as well.

## 2. Materials and Methods

### 2.1. Chemicals

Capsaicin standard, Mueller-Hinton broth (MHB), Sabouraud Dextrose Agar (SDA), RPMI (Roswell Park Memorial Institute) 1640 medium, and Tryptic Soy Agar (TSA) were purchased from Sigma-Aldrich (Munich, German). All other chemicals and reagents used were of analytical grade.

### 2.2. Plant Material and Extraction Procedure

*Capsicum* annuum fruits were purchased from the trade of Chieti, Abruzzo, Italy.

The plant material was identified at the Department of Pharmacy, Faculty of Pharmacy, University of Chieti in Pescara, Italy, where the voucher specimens have been deposited. During the experimental phase, the extract was prepared as follows: Oven-dried and pulverized plant material (2.5 g) was extracted by maceration process as described elsewhere with 98% ethanol (25 mL) for 24 h [25]. The extract was then filtered through Whatman No. 1 paper and concentrated under vacuum. The dried extract was taken in acetonitrile (5 mL) and left for 12 h under magnetic stirring. The extraction yields are given in Table 1.

### 2.3. HPLC Analyses

The concentration of the capsaicin in the obtained extract was analyzed by using a slightly modified method as described by Abdullah Al Othman Z. et al. [25]. HPLC apparatus was a Waters 600 HPLC pump (Waters Corporation, Milford, MA, USA), equipped with a Waters 2996 photodiode array detector. The mobile phase, consisting in a mixture 50:50 of water and acetonitrile, was flushed in a Hyperclone C18 column (250 × 4.60 mm, 5 µm) at a flow rate of 1 mL/min. The injection volume was 10 μL. All chromatographic experiments were performed at room temperature. Capsaicin content in chili pepper extracts were recorded at a wavelength of 228 nm. Capsaicin was identified based on its chromatographic retention time and quantified by comparing integrated peak areas to calibration curves obtained with analytical standards prepared in a concentration range from 0.01–0.5 mg/mL [26]. Results were expressed in mg per g of pulverized plant material. All HPLC analyses were performed in triplicate.

### 2.4. Formulation with Capsaicin Extract

The carbopol (0.08 g) was finely dispersed in water and continuously stirred at 350 rpm until a homogeneous dispersion was obtained. Then, propylene glycol (0.5 g), sodium Ethylenediaminetetraacetic acid (EDTA) (0.02 g), and the capsaicin extract dissolved in ethanol (2.495 g) were added to the Carbopol mixture. The extract contained a total amount of capsaicin of 2.5 mg. Sodium hydroxide (0.11 g) was added to the dispersion to afford the gel.

### 2.5. Evaluation of Rheological Properties

Rheological characterization was achieved with a Haake M.A.R.S. II Thermo Scientific modular rheometer consisting of a cone-plate system (60 mm diameter, and 0.5° cone angle), a Haake Phoenix Thermo Electron Corporation thermo-controller system to the temperature regulation, and RheoWin software 3.61 (Thermo Fisher Scientific, Waltham, MA, USA) for data elaboration. Static and oscillatory tests were performed at the fixed temperature of 34 ± 0.5 °C. The influence of the viscosity of the gel was investigated through flow curves obtained varying D from 0.1 to 100 s^−1^. In the creep and recovery tests, the resulting samples deformation (γ) was measured over time (t) when a constant shear stress (τ), ranging from 2 to 12 Pa, was applied and then removed for a fixed period. Dynamic tests were carried out to estimate both the storage (G′) and the loss (G″) module. Oscillation frequency sweep analysis was carried out from 0.1 to 10 Hz, employing a shear stress of 3 Pa, identified through the oscillation stress sweep test, satisfying the linear viscoelasticity [27,28].

### 2.6. In Vitro Release of Capsaicin from Carbopol-Based Formulation

In vitro release studies were performed on an incubated capsaicin-Carbopol formulation using a dialyses membrane method. Experiments were performed at 34 °C and under magnetic stirring (250 rpm). To quantify the amount of released capsaicin, at fixed time points, aliquots of 1 mL were withdrawn, from the acceptor compartment, and analyzed by HPLC, using the chromatographic condition previously reported. After each withdraws the medium was replaced with an identical volume of fresh medium. The acceptor compartment consisted of water enriched with 1% of Cremophor to ensure the complete capsaicin solubilisation in the aqueous medium.

### 2.7. Antimicrobial Susceptibility Testing

In vitro antimicrobial activity of Capsaicin and Carbopol formulation were assessed against four bacterial strains, namely *Escherichia coli* (ATCC 10536), *Bacillus cereus* (Peru MycA 4), *Salmonella typhi* (Peru Myc 7), *Staphylococcus aureus* (ATCC 6538); and four yeasts namely *Candida tropicalis* (YEPGA 6184), *Candida albicans* (YEPGA 6379), *Candida parapsilosis* (YEPG 6551) and *Candida albicans* (YEPG 6138). 

*Candida parapsilosis* (ATCC 22019) and *Candida krusei* (ATCC 6258) strains were used as quality controls for antifungal assays (CLSI 2008; CLSI 2012; CLSI 2017). 

Voucher cultures are maintained in the PeruMycA culture collection of the Department of Chemistry, Biology and Biotechnology (DCBB) (University of Perugia, Italy) and are available upon request. 

For Minimum Inhibitory Concentration (MIC) determination, Capsaicin and Carbopol formulation were used in the range 0.312–10 µg/mL.

### 2.8. Antibacterial Activity Assay

The Minimum inhibitory concentrations (MICs) of Capsaicin and Carbopol formulation were determined according to the broth microdilution method of the Clinical and Laboratory Standards Institute, M07-A10 [29]. 

Briefly, bacterial suspension (inocula) for MIC determination was prepared as follows: a few colonies from 24 h-old cultures on Tryptic Soy Agar (TSA) plates were transferred in Mueller-Hinton broth (MHB) and statically incubated overnight at 37 °C. Cell density of each inoculum was hence adjusted according to the opacimetric standard Mac Farland 0.5 (1.5 × 10^8^ CFU/mL). This was confirmed by plating serial dilutions of the inoculum suspensions on Mueller- Hinton Agar (MHA). 20 μL of bacterial suspensions were used to inoculate 1 mL of MHB medium containing serial dilutions of active Capsaicin or Carbopol formulation. 

The set-up included bacterial growth controls in wells containing 10 μL of the test inoculum and negative controls without bacterial inoculum. MIC end-points were determined after 18–20 h incubation in ambient air at 35 °C [30]. MIC end-points were defined as the lowest concentration of Capsaicin, Carbopol formulation and ciprofloxacin that totally inhibited bacterial growth [30]. Each test was done in triplicate. Geometric means and MIC ranges were calculated.

### 2.9. Antifungal Activity Assay 

Susceptibility testing against yeasts, was performed according to the CLSI M27-4th ed. [31] and M60 [32] protocols.

RPMI (Roswell Park Memorial Institute) 1640 medium (Sigma) with L-glutamine and without sodium bicarbonate, supplemented with 2% glucose (*w*/*v*), buffered with 0.165 M morpholinepropanesulphonic acid (MOPS), pH 7.0, was used throughout the study. 

The inoculum suspensions of yeasts were prepared from 7-day-old cultures grown on Sabouraud Dextrose Agar (SDA; Difco) at 25 °C and adjusted spectrophotometrically to optical densities that ranged from 0.09 to 0.11 (Mac Farland standard). The inoculum suspensions were diluted to a ratio of 1:50 in RPMI 1640 to obtain twice an inoculum size ranging from 0.2 to 0.4 × 10^4–5^ CFU/mL. This was further confirmed by plating serial dilutions of the inoculum suspensions on SDA.

MIC end-points (µg/mL) were determined after 24 h (for yeasts) of incubation in ambient air at 30 °C [31,32].

For the Capsaicin and Carbopol formulation, the MIC end-points were defined as the lowest concentration that showed total growth inhibition [33]. The MIC end-point for fluconazole was defined as the lowest concentration that inhibited 50% of the growth when compared with the growth control [31,32].

Geometric means and MIC ranges were determined from the three biological replicates to allow comparisons between the activities of Capsaicin and Carbopol formulation.

### 2.10. Statistical Analysis

The data analysis was performed using Microsoft Excel 2007 software, GraphPad Prism 8 and MedCalc for Windows (version 12.5.0.0.). Correlations were assessed using Pearson’s correlation coefficient (r) and *p* < 0.05 was considered statistically significant.

## 3. Results and Discussion

### 3.1. Content of Capsaicin in Ethanolic Extracts and Carbopol Gel Formulation

For centuries, capsaicin was used unknowingly in the form of chili peppers in foods in order to enhance their taste, aroma, color and hotness [34]. Besides it was used in the food industry, capsaicin has found its application in pharmaceutical industry as well providing many health benefits and treatment strategies for medical conditions [35].

Pepper varieties from *Capsicum annuum*, *C. frutescens* and *C. chinense* were found to contain 0.22–20 mg total capsaicinoids/g of dry weight [36]. In another study, cayenne pepper samples had mean capsaicin and dihydrocapsaicin contents of 1.32 and 0.83 mg/g dry weight, respectively [37].

Many analytical methods, such as HPLC, HPTLC, RP-HPLC, are suggested in years by many scientists for identification and quantification of Capsaicin from fruits of chili peppers [24,38,39]. 

In this study, we have proposed a slight modified HPLC method for the determination of capsaicin from ethanolic extracts and from pharmaceutical topical formulations. The ethanolic extract, obtained by maceration, was characterized for the capsaicin content by HPLC and then used for the gel preparation (Table 1).

### 3.2. Characterization of Capsaicin Formulations

The Carbopol-based formulations containing capsaicin were subjected to rheological studies to investigate mucoadhesive properties of the gel. As confirmed by both static and oscillatory tests, a structured system, characterized by a viscoelastic response, was produced. Figure 1 shows the influence that the shear rate (D) has on the viscosity: a flow curves typical of a shear thinning pseudo-plastic material, for which the apparent viscosity (η) drops with the D increasing. The creep and recovery analyses were employed as additional test to estimate the gel resistance once subjected to mechanical stresses, ranging from 2 to 12 Pa, included in the shear stress interval satisfying the linear viscoelastic range (LVR) [40]. Figure 2, plotting sample strain (ɣ) against time for the different shear stresses applied, revealed a typical viscoelastic behavior. The reported graphs were characterized during the creep cycle (stress application) by an initial deformation, which followed after stress removing and by a partial recovery and an irreversible deformation, due to its elastic and viscous component, respectively [41]. In the unloaded (data not shown) and in capsaicin-loaded gels, comparable deformations that were directly proportional to the applied forces were detected, suggesting that the addition of the drug did not significantly influence the intramolecular association in the gel network.

The measurements conduced in the oscillation mode are depicted in Figure 3. Oscillation frequencies sweep test was carried out in a frequency range from 0.1–10 Hz, applying a shear stress included in the LVR. Results suggest that, both dynamic modules are not frequency-dependent.

On the contrary, the complex viscosity (η*) gradually declines with growing frequencies, further supporting the shear-thinning behavior predicted with the flow curve [42]. 

Moreover, for all considered frequencies, the G’ modulus is greater than the G" modulus indicating that strong interconnections are established among polymer chains. 

Taking together data obtained from rheological studies, we can assume that the formulation containing capsaicin is a well-structured gel.

### 3.3. In Vitro Release Assay

In vitro release studies were performed evaluating the percentage of capsaicin released from the Carbopol-based formulation over time (Figure 4). A plateau region was observed after 40 h. The maximum percentage of capsaicin released was 47%, suggesting that the gel significantly affects the entrapped drug movement causing its retention in its network [43]. These results could be indicative of a high capsaicin affinity for the gel constituents and strong interactions could be supposed to be responsible for the low propensity of this compound to move in the acceptor aqueous medium.

### 3.4. Antimicrobial Properties

Finally, antimicrobial properties were conducted with the aim to explore the influence of Carbopol-based formulation on capsaicin antimicrobial effect. As depicted in Table 2, Figure 5 and Figure 6 the Carbopol formulation increased the antibacterial and antifungal effect of capsaicin, with a MIC value reduction of at least 50%. Considering the findings of Scalzo and colleagues [44], a synergistic effect between capsaicin and Carbopol is hypothesized regarding the antimicrobial effects, thus suggesting potential applications in cosmetics.

## 4. Conclusions

This paper presents the results of a good form of hydrogel formulation of capsaicin extract. The findings about the inner structure of the gel conclude for a pseudo-plastic form with good rheological properties. 

The in vitro evaluation of drug release suggests a prolong release and good bioavailability of this formulation and may have great relevance in the local treatment of inflammatory joints diseases. Wang et al. (2001) evaluated the skin absorption of capsaicin from different hydrogels and we may conclude that the in vitro permeation of capsaicin from hydrogels depends on the physicochemical nature and the concentration of the polymer used. They emphasize that higher levels of capsaicin are released from anionic polymer hydrogels than cream bases [45]. 

From the antimicrobial and antifungal properties assessment, we may conclude that our formulation has good antimicrobial effects against *Escherichia coli* and the same effects between *Bacillus cereus* and *Salmonella typhi* compare to pure capsaicin. Also, the Carbopol formulation exhibits good antifungal properties from each species of *Candida*, especially from *Candida albicans*. 

According these results, we strongly recommend this formulation for dermatological use due to its anti-inflammatory and antimicrobial properties.

## Figures and Tables

**Figure 1 biomolecules-11-00432-f001:**
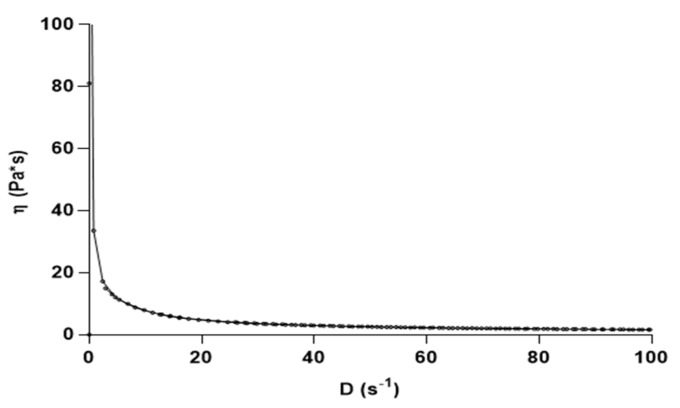
Apparent viscosity as function of shear rate in Carbopol-based formulation.

**Figure 2 biomolecules-11-00432-f002:**
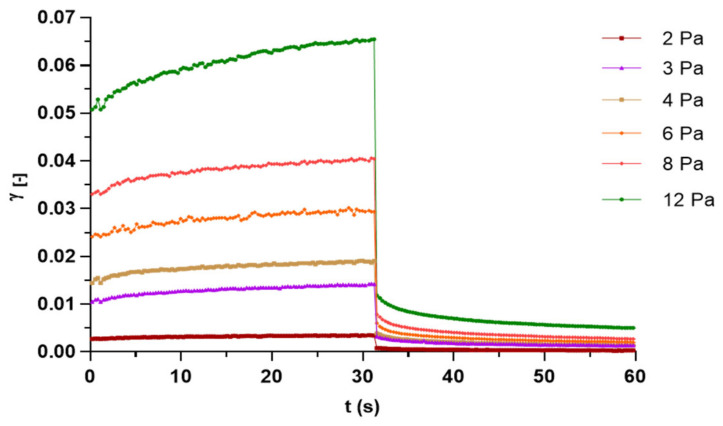
Creep and recovery test in Carbopol-based formulation containing capsaicin at different shear stress.

**Figure 3 biomolecules-11-00432-f003:**
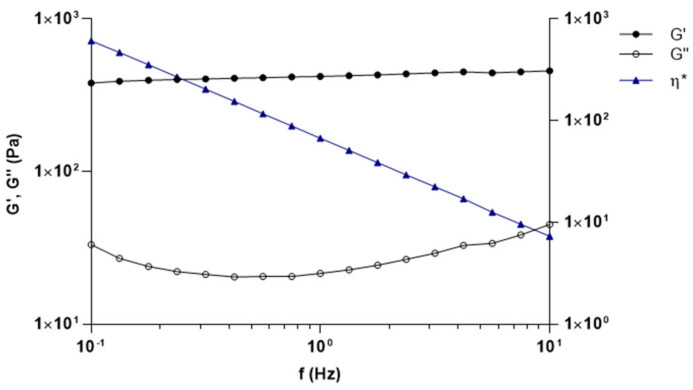
G’ and G" module and dynamic viscosity as a function of frequencies in Carbopol-based formulation containing capsaicin.

**Figure 4 biomolecules-11-00432-f004:**
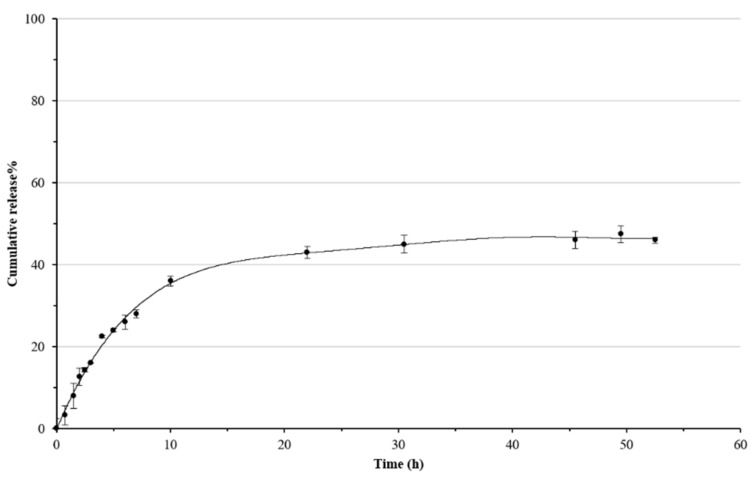
Cumulative release of capsaicin from Carbopol-based formulation.

**Figure 5 biomolecules-11-00432-f005:**
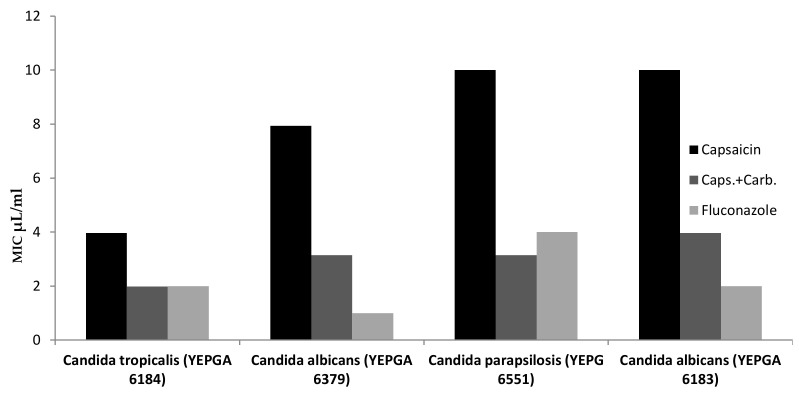
Fungal Minimum Inhibitory Concentration (MIC) µL/mL of Capsaicin from Carbopol-based formulation.

**Figure 6 biomolecules-11-00432-f006:**
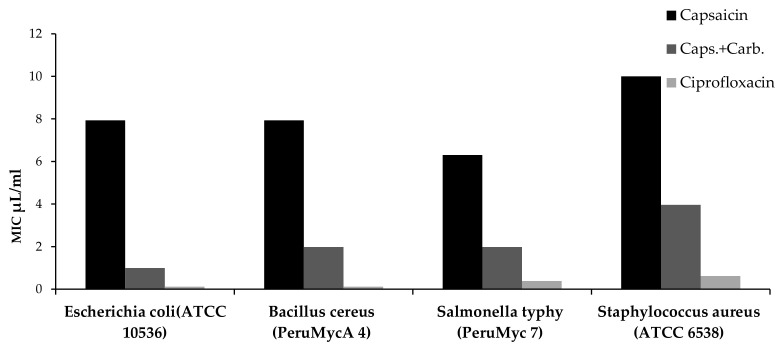
Bacterial Minimum Inhibitory Concentration (MIC) µL/mL of Capsaicin from Carbopol-based formulation.

**Table 1 biomolecules-11-00432-t001:** HPLC quantification of Capsaicin from extracts and extraction yield.

Pepper Type	Capsaicin (mg/g) ^1^	Yield ^2^
*Caspicum* annuum	2.109 ± 0.028	0.21%

^1^ Content determined by HPLC and expressed as mg/g of vegetable material; ^2^ Extract yield expressed as percentage weight of air-dried plant material.

**Table 2 biomolecules-11-00432-t002:** Minimum inhibitory concentration of Capsaicin and Carbopol formulation.

Microrganisms (ID)	Minimum Inhibitory Concentration (MIC) µL/mL		
Yeasts	Capsaicina	Caps.+Carb.	Fluconazole
*Candida tropicalis* (YEPGA 6184)	3.968 (2.5–5)	1.984 (1.25–2.5)	2
*Candida albicans* (YEPGA 6379)	7.937 (5–10)	3.149 (2.5–5)	1
*Candida parapsilosis* (YEPG 6551)	>10	3.149 (2.5–5)	4
*Candida albicans* (YEPGA 6183)	>10	3.968 (2.5–5)	2
**Bacteria**			**Ciprofloxacin**
*Escherichia coli* (ATCC 10536)	7.937 (5–10)	0.992 (0.625–1.25)	<0.12
*Bacillus cereus* (Peru MycA 4)	7.937 (5–10)	1.984 (1.25–2.5)	<0.12
*Salmonella typhy* (Peru Myc 7)	6.299 (5–10)	1.984 (1.25–2.5)	0.38 (0.49–0.24)
*Staphylococcus aureus* (ATCC 6538)	>10	3.968 (2.5–5)	0.62 (0.98–0.49)

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
