# Peer review of "Evaluation of In Vitro Capsaicin Release and Antimicrobial Properties of Topical Pharmaceutical Formulation"

_biomolecules, 2021, doi:10.3390/biom11030432_

Round 1
Reviewer 1 Report
The manuscript entitled “Evaluation of in vitro capsaicin release and antimicrobial properties of topical pharmaceutical formulation” have demonstrated formulation of topical capsaicin ethanolic extract in pharmaceutical gels with prolonged effect avoiding its hepatic metabolism and improving bioavailability. Moreover, antibacterial and antifungal effects against E.coli, B. cereus, S. typhy and S. aureus and Candida sp. were evaluated.
Authors should correct manuscript according to the suggestion.
Minor issues:
Author should corrected microorganisms names in whole manuscript (names should be italicized) for example Line 129 – 130,
Introduction
Introduction should be rewritten; authors should describe background for antimicrobial activity of capsaicin, give some examples
Material and methods
Line 83 – 84: “Chemicals” should be described before “Plant Material and Extraction Procedure”
Line 128 – 131: Antimicrobial activity should be described in details (inoculum size, medium, incubation conditions and concentration of capsaicin)
Line 131: Authors should added positive controls for antibacterial and antifungal assay.
Results and discussion
Line 149 – 152: Authors should described advantages of modified HPLC methods for the determination of capsaicin; some advantages compare with other method
Line 191: “antifungal” will be better than “mycostatic”
Figure 5 and 6, Table 2: positive control are missing
Author Response
Reviewer 1:
- Author should corrected microorganisms names in whole manuscript (names should be italicized) for example Line 129 – 130,
Response to Reviewer 1: The corrections have been carried out as follows:
Antimicrobial susceptibility testing
In vitro antimicrobial activity of Capsaicin and Carbopol formulation were assessed against four bacterial strains, namely Escherichia coli (ATCC 10536), Bacillus cereus (Peru MycA 4), Salmonella typhi (Peru Myc 7), Staphylococcus aureus (ATCC 6538); and four yeasts namely Candida tropicalis (YEPGA 6184), Candida albicans (YEPGA 6379), Candida parapsilosis (YEPG 6551) and Candida albicans (YEPG 6138).
Candida parapsilosis (ATCC 22019) and Candida krusei (ATCC 6258) strains were used as quality controls for antifungal assays (CLSI 2008; CLSI 2012; CLSI 2017).
Voucher cultures are maintained in the PeruMycA culture collection of the Department of Chemistry, Biology and Biotechnology (DCBB) (University of Perugia, Italy) and are available upon request.
For Minimum Inhibitory Concentration (MIC) determination, Capsaicin and Carbopol formulation were used in the range 0.312–10 µg mL-1.
Reviewer 1
Introduction
- Introduction should be rewritten; authors should describe background for antimicrobial activity of capsaicin, give some examples
Response to Reviewer 1: The corrections have been carried out as follows:
Beside its multiple pharmacological properties, capsaicin has recently also attracted attention for its antimicrobial activities (Luo et al. 2011). Currently, researchers have shown that capsaicin can inhibit the growth of some food-borne pathogenic microorganisms, such as Listeria monocytogenes, Helicobacter pylory, Pseudomonas aeruginosa, Botrytis cinerea, Aspergillus niger and Penicillium expansum (Cowan 1999, Dorantes et al. 2000, Xing et al. 2006, Omolo et al 2014, Tayseer et al. 2020), whereas an anti-virulence activity has been demonstrated against Vibrio cholerae, Staphylococcus aureus and Poephyromonas gingivalis (Chatterjee et al. 2010, Kalia et al. 2012, Qiu et al 2012, Zhou et al. 2014).
In a study by Nascimento and colleagues (2014), it has been tested the antimicrobial activity of capsaicin against Gram negative bacteria (Escherichia coli, Pseudomonas aeruginosa, and Klebsiella pneumoniae), Gram positive bacteria (Enterococcus faecalis, Bacillus subtilis, and Staphylococcus aureus) and one yeast (Candida albicans). MIC values obtained ranging between 0.06 and 25 μg mL-1 (Nascimento et al. 2014).
Marini et al. (2015) reported that the minimal inhibitory concentrations (MICs) for capsaicin to inhibit erythromycin-resistant and erythromycin-susceptible strains were prevalently in a narrow range (0.064-0.128 mg mL-1), where the most common MIC was 0.128 mg mL-1 and the highest MIC was exhibited at 0.512 mg mL-1 (Marini et al. 2015). Moreover, Ndashe et al. (2020) showed that the minimum bactericidal concentration (MBC) of capsaicin on the growth of Lactococcus garvieae was 0.1967mg mL-1, according with other studies (Fieira et al., 2013; Marini et al., 2015).
Reviewer 1:
Material and methods
- Line 83 – 84: “Chemicals” should be described before “Plant Material and Extraction Procedure”
Response to Reviewer 1: The corrections have been carried out as follows:
- Materials and Methods
Chemicals
Capsaicin standard, Mueller-Hinton broth (MHB), Sabouraud Dextrose Agar (SDA), RPMI (Roswell Park Memorial Institute) 1640 medium, and Tryptic Soy Agar (TSA) were purchased from Sigma-Aldrich. All other chemicals and reagents used were of analytical grade.
- Line 128 – 131: Antimicrobial activity should be described in details (inoculum size, medium, incubation conditions and concentration of capsaicin)
- Line 131: Authors should added positive controls for antibacterial and antifungal assay.
Antimicrobial susceptibility testing
In vitro antimicrobial activity of Capsaicin and Carbopol formulation were assessed against four bacterial strains, namely Escherichia coli (ATCC 10536), Bacillus cereus (Peru MycA 4), Salmonella typhi (Peru Myc 7), Staphylococcus aureus (ATCC 6538); and four yeasts namely Candida tropicalis (YEPGA 6184), Candida albicans (YEPGA 6379), Candida parapsilosis (YEPG 6551) and Candida albicans (YEPG 6138).
Candida parapsilosis (ATCC 22019) and Candida krusei (ATCC 6258) strains were used as quality controls for antifungal assays (CLSI 2008; CLSI 2012; CLSI 2017).
Voucher cultures are maintained in the PeruMycA culture collection of the Department of Chemistry, Biology and Biotechnology (DCBB) (University of Perugia, Italy) and are available upon request.
For Minimum Inhibitory Concentration (MIC) determination, Capsaicin and Carbopol formulation were used in the range 0.312–10 µg mL-1.
Antibacterial activity assay
The Minimum inhibitory concentrations (MICs) of Capsaicin and Carbopol formulation were determined according to the broth microdilution method of the Clinical and Laboratory Standards Institute, M07-A10 (CLSI 2015).
Briefly, bacterial suspension (inocula) for MIC determination was prepared as follows: a few colonies from 24 h-old cultures on Tryptic Soy Agar (TSA) plates were transferred in Mueller-Hinton broth (MHB) and statically incubated overnight at 37 °C. Cell density of each inoculum was hence adjusted according to the opacimetric standard Mac Farland 0.5 (1.5× 108 CFU/ml). This was confirmed by plating serial dilutions of the inoculum suspensions on Mueller- Hinton Agar (MHA). 20 μl of bacterial suspensions were used to inoculate 1 ml of MHB medium containing serial dilutions of active Capsaicin or Carbopol formulation.
The set-up included bacterial growth controls in wells containing 10 μL of the test inoculum and negative controls without bacterial inoculum. MIC end-points were determined after 18-20 hours incubation in ambient air at 35°C (Angelini et al. 2020). MIC end-points were defined as the lowest concentration of Capsaicin, Carbopol formulation and ciprofloxacin that totally inhibited bacterial growth (Angelini et al. 2020). Each test was done in triplicate. Geometric means and MIC ranges were calculated.
Antifungal activity assay
Susceptibility testing against yeasts, was performed according to the CLSI M27-4th ed. (CLSI 2017a) and M60 (CLSI 2017b) protocols.
RPMI (Roswell Park Memorial Institute) 1640 medium (Sigma) with L-glutamine and without sodium bicarbonate, supplemented with 2% glucose (w⁄v), buffered with 0.165 mol l-1 morpholinepropanesulphonic acid (MOPS), pH 7.0, was used throughout the study.
The inoculum suspensions of yeasts were prepared from 7-day-old cultures grown on Sabouraud Dextrose Agar (SDA; Difco) at 25 °C and adjusted spectrophotometrically to optical densities that ranged from 0.09 to 0.11 (Mac Farland standard). The inoculum suspensions were diluted to a ratio of 1:50 in RPMI 1640 to obtain twice an inoculum size ranging from 0.2 to 0.4 × 104–5 CFU mL-1. This was further confirmed by plating serial dilutions of the inoculum suspensions on SDA.
MIC end-points (µg mL-1) were determined after 24 hours (for yeasts) of incubation in ambient air at 30°C (CLSI 2017a, b).
For the Capsaicin and Carbopol formulation, the MIC end-points were defined as the lowest concentration that showed total growth inhibition (Pagiotti et al. 2011). The MIC end-point for fluconazole was defined as the lowest concentration that inhibited 50% of the growth when compared with the growth control (CLSI 2017 a,b)
Geometric means and MIC ranges were determined from the three biological replicates to allow comparisons between the activities of Capsaicin and Carbopol formulation.
Reviewer 1:
Results and discussion
- Line 149 – 152: Authors should described advantages of modified HPLC methods for the determination of capsaicin; some advantages compare with other method
Response to Reviewer 1: The authors just modified the flow rate compared to the method reported by Abdullah Al Othman Z. et al. to take advantage of higher accuracy for the capsaicin detection. Moreover, with the decrease of the flow rate, the capsaicin retention time was increased, and the peak separation improved.
Reviewer 1:
- Line 191: “antifungal” will be better than “mycostatic”
Response to Reviewer 1: As depicted in table 2, figure 5 and 6 the Carbopol formulation increased the antibacterial and antifungal effect of capsaicin, with a MIC value reduction of at least 50%.
Reviewer 1:
- Figure 5 and 6, Table 2: positive control are missing
Response to Reviewer 1: The suggested corrections have been carried in the manuscript as follows:
Table 2: Minimum Inhibitory Concentration of Capsaicin and Carbopol formulation
|
Microrganisms (ID) |
Minimum Inhibitory Concentration (MIC) µl/mL |
|||
|
Capsaicina |
|
Caps.+Carb. |
Fluconazole |
|
|
Yeasts |
|
|
||
|
Candida tropicalis (YEPGA 6184) |
3.968 (2.5-5) |
|
1.984 (1.25-2.5) |
2 |
|
Candida albicans (YEPGA 6379) |
7.937 (5-10) |
|
3.149 (2.5-5) |
1 |
|
Candida parapsilosis (YEPG 6551) |
>10 |
|
3.149 (2.5-5) |
4 |
|
Candida albicans (YEPGA 6183) |
>10 |
|
3.968 (2.5-5) |
2 |
|
Bacteria |
|
Ciprofloxacin |
||
|
Escherichia coli (ATCC 10536) |
7.937 (5-10) |
|
0.992 (0.625-1.25) |
<0.12 |
|
Bacillus cereus (Peru MycA 4) |
7.937 (5-10) |
|
1.984 (1.25-2.5) |
<0.12 |
|
Salmonella typhy (Peru Myc 7) |
6.299 (5-10) |
|
1.984 (1.25-2.5) |
0.38 (0.49–0.24) |
|
Staphylococcus aureus (ATCC 6538) |
>10 |
|
3.968 (2.5-5) |
0.62 (0.98–0.49) |
Figure 5. Fungal Minimum Inhibitory Concentration (MIC) µl/mL of Capsaicin from Carbopol-based formulation.
Figure 6. Bacterial Minimum Inhibitory Concentration (MIC) µl/mL of Capsaicin from Carbopol-based formulation.

Reviewer 2 Report
The authors have developed a new pharmaceutical formulation with capsaicin and present its rheological and antimicrobial studies. Both the extraction and the HPLC analysis of C from different sources have already been extensively studied. The study of this new formulation is serious and rigorous but it could only be accepted for publication if the improvement of this hydrogel with respect to other already published formulations is highlighted.
On the other hand, the references do not follow the order in the text. It gives the feeling that as in this publication the materials and methods are ahead of the results, the authors have not been careful enough to change journal publication.
Author Response
Reviewer 2:
- On the other hand, the references do not follow the order in the text. It gives the feeling that as in this publication the materials and methods are ahead of the results, the authors have not been careful enough to change journal publication
Response to Reviewer 2: The suggested corrections have been carried in the manuscript as follows:
References:
- Babbar S, Marier JF, Mouksassi MS, Beliveau M, Vanhove GF, Chanda S, Bley K. Pharmacokinetic analysis of capsaicin after topical administration of a high-concentration capsaicin patch to patients with peripheral neuropathic pain. Ther Drug Monit.2009 Aug; 31(4):502-10
- Peng X., Wen X., Pan X., Wang R., Chen B. and Wu C. Design and In Vitro Evaluation of Capsaicin Transdermal Controlled Release Cubic Phase Gels. AAPS PharmSciTech, 2010, Vol. 11, No. 3, 1405-1410.
- Hayman M, Kam PCA. Capsaicin: a review of its pharmacology and clinical applications. CurrAnaesthCrit Care. 2008;19 (5–6):338–43.
- Kosuge, S.; Furuta, M. Studies on the pungent principle of Capsicum. Part XIV: Chemical constitution of the pungent principle. Agric. Biol. Chem. 1970, 34, 248-256.
- Schumacher M, Pasvankas G Topical capsaicin formulations in the management of neuropathic pain. Prog Drug Res.2014; 68:105-28.
- Derry S, Lloyd R, Moore RA, McQuay HJ. Topical capsaicin for chronic neuropathic pain in adults. Cochrane Database Syst Rev.2009 Oct 7;(4)
- Arora R., Gill N.S, Chauhan G., Rana A.C. An Overview about Versatile Molecule Capsaicin. IJPSDR. 2011; 3(4): 280-286
- Luo XJ, Peng J, Li YJ. Recent advances in the study on capsaicinoids and capsinoids. Eur J Pharmacol 2011, 650:1-7.
- Cowan MM. Plant products as antimicrobial agents. Clin. Microbiol. Rev. 12:564-582.
- Dorantes L, Colmenero R, Hernandez H, Mota L, Jaramillo ME, Fernandez E, Solano C (2000) Inhibition of growth of some foodborne pathogenic bacteria by Capsicum annuum extracts. Int J Food Microbiol 1999, 57:125–128
- Xing F, Cheng G, Yi K. 2006. Study on the antimicrobial activities of the capsaicin microcapsules. Journal of Applied Polymer Science, 2006, 102(2):1318-1321.
- Omolo MA, Wong ZZ, Mergen AK, Hastings JC, Le NC, Reiland HA et al. Antimicrobial properties of chili peppers. J Infect Dis Ther 2014, 2:1-8.
- Tayseer I, Aburjai T, Abu-Qatouseh L, AL-Karabieh N, Ahmed W, Al-Samydai A. In vitro Anti-Helicobacter pylori Activity of Capsaicin. J Pure Applied Microbiology 2020, 14 (1): 279-286
- Chatterjee S., Asakura M., Chowdhury N., Neogy SB, Sugimoto N., Haldar S., et al. Capsaicin, a potential inhibitor of cholera toxin production in Vibrio cholerae. FEMS Microbiol. Let. 2010, 306:54-60.
- Kalia NP, Mahajan P, Mehera R, Nargotra A, Sharma JP, Koul S et al. (2012) Capsaicin, a novel inhibitor of the NorA efflux pump, reduces the intracellular invasion of Staphylococcus aureus. J Antimicrob Chemother 2012, 67:2401-2408.
- Qiu J, Niu X, Wang J, Xing Y, Leng B, Dong J et al. Capsaicin protects mice from community-associated methicillin-resistant Staphylococcus aureus pneumonis. PLoS ONE 2012, 7:e33032.
- Zhou Y, Guan X, Zhu W, Liu Z, Wang X, Yu H. et al. Capsaicin inhibits Porphyromonas gingivalis growth, biofilm formation, gingivomucosal inflammatory cytokine secretion, and in vitro osteoclastogenesis. Eur. J. Clin. Microbiol. Infect. Dis. 2014, 33:211-219.
- Nascimento PLA, Nascimento TCES, Ramos NSM, Silva GR, Gomes JEG, Falcão REA, Moreira KA, Porto ALF, Silva TMS. Quantification, Antioxidant and Antimicrobial Activity of Phenolics Isolated from Different Extracts of Capsicum frutescens (Pimenta Malagueta). Molecules 2014, 19:5434-5447.
- Marini E, Magi G, Mingoia M, Pugnaloni A, Facinelli B (2015) Antimicrobial and Anti-Virulence Activity of Capsaicin Against Erythromycin-Resistant, Cell-Invasive Group A Streptococci. Front Microbiol 2015, 6, https://doi.org/10.3389/fmicb.2015.01281.
- Fieira C, Oliveira F, Calegari R., Machado A, Coelho AR. 2013. Inibidores naturais no controle in vitro e in vivo de Penicillium expansum. Food Sci Technol 2013, 33, 40–46. https://doi.org/10.1590/S0101-20612013000500007
- Ndashe K, Ngh’ake S, Pola E, Sakala EM, Kabwali E, Moonga L, Kefi A.S., Hang’ombe B.M.The Potential of Capsicum annum Extracts to Prevent Lactococcosis in Tilapia. BioRxiv 2020.06.09.142406; doi: https://doi.org/10.1101/2020.06.09.142406
- Farmacopea Ufficiale della Repubblica Italiana – Droghe Vegetali e Preparazioni, Roma 1991; 100-105.
- Wagnerit H., Bladt S. Plant drug analysis, A Thin Layer Chromatography Atlas, 2nd ed.; Springer-Verlag Berlin Heidelberg New York, 2001; pp. 289-292.
- Goci E., Haloçi E., Vide K., Malaj L. Application and comparison of three different extraction methods of capsaicin from capsicum fruits. AJPhSci.2013Vol. 1 No. 1
- Abdullah Al Othman Z., Ahmed Y., Habila M., and Ghafar A. Determination of Capsaicin and Dihydrocapsaicin in Capsicum Fruit Samples using High Performance Liquid Chromatography. Molecules 2011, 16, 8919-8929
- Soetarno S, Sukrasn O, Yulinah E. Antimicrobial activities of the ethanol extracts of capsicum fruits with different levels of pungency. JMS. 1997; 2 (2): 57-63.
- Di Stefano, A., D’Aurizio, E., Trubiani, O., Grande, R., Di Campli, E., Di Giulio, M., Di Bartolomeo, S., Sozio, P., Iannitelli, A., Nostro, A., Cellini, L., Viscoelastic properties of Staphylococcus aureus and Staphylococcus epidermidis mono-microbial biofilms. Microb. Biotechnol. 2009.2, 634–641.
- Iannitelli, A., Grande, R., Di Stefano, A., Di Giulio, M., Sozio, P., Bessa, L.J., Laserra, S., Paolini, C., Protasi, F., Cellini, L., Potential antibacterial activity of carvacrol-loaded poly (DL-lactide-co-glycolide) (PLGA) nanoparticles against microbial biofilm. Int. J. Mol. Sci. 2011.12, 5039–5051.
- Methods for Dilution Antimicrobial Susceptibility Tests for Bacteria That Grow Aerobically. Approved Standard-10th ed. CLSI document M07-A10. Clinical and Laboratory Standards Institute, 950 West Valley Road, Suite 2500, Wayne, Pennsylvania 19087, USA, 2015.
- Angelini, P.; Venanzoni, R.; Angeles Flores, G.; Tirillini, B.; Orlando, G.; Recinella, L.; Chiavaroli, A.; Brunetti, L.; Leone, S.; Di Simone, S.C.;Ciferri, M.C.;Zengin, G.;Ak, G.;Menghini, L.; Ferrante, C. Evaluation of Antioxidant, Antimicrobial and Tyrosinase Inhibitory Activities of Extracts from Tricholosporum goniospermum, an Edible Wild Mushroom. Antibiotics (Basel). 2020, 9(8). doi: 10.3390/antibiotics9080513.
- Reference Method for Broth Dilution Antifungal Susceptibility Testing of Yeast. 4th ed. CLSI standard M27. Clinical and Laboratory Standards Institute, 950 West Valley Road, Suite 2500, Wayne, Pennsylvania 19087, USA, 2017a.
- Performance Standards for Antifungal Susceptibility Testing of yeasts. 1st ed. CLSI supplement M60. Clinical and Laboratory Standards Institute, 950 West Valley Road, Suite 2500, Wayne, Pennsylvania 19087, USA, 2017b.
- Pagiotti R, Angelini P, Rubini A, Tirillini B, Granetti B, Venanzoni R. Identification and characterisation of human pathogenic filamentous fungi and susceptibility to Thymus schimperi essential oil. Mycoses 2011, 54(5):e364-76. DOI: 10.1111/j.1439-0507.2010.01926.x.
- Goodwin, D. C., M. Hertwig. Peroxidase-catalyzed oxidation of capsaicinoids: steady-state and transient state kinetic studies. Arch.Biochem. Biophys. 2003, 417: 18–26.
- Szolcsanyi, J. Forty years in capsaicin research for sensory pharmacology and physiology. Neuropeptides. 2004, 38: 377-384.
- Thomas, B.V.; Schreiber, A.A.; Weisskopf, C.P. Simple method for quantitation of capsaicinoids in peppers using capillary gas chromatography. J. Agric. Food Chem. 1998, 46, 2655-2663.
- Lopez-Hernandez, J.; Oruna-Concha, M.J.; Simal-Lozano, J.; Gonzales-Castro, M.J.; Vazquez-Blanco, M.E. Determination of capsaicin and dihydrocapsaicin in cayenne pepper and padron peppers by HPLC. Dtsch. Lebensmitt. Rundsch.1996, 92, 393-395.
- Cheema SK, Pant MR. Estimation of capsaicin in seven cultivated varieties of Capsicum Annuum L. Res. J. Pharm. Biol. Chem. Sci.2011, 2(2), 701.
- Zahra N, Nisa A, at al. Estimation of capsaicin in different chili varieties using different Extraction Techniques and HPLC method: A Review. Pak. J. Food Sci., 2016, 26(1): 54-60.
- Chytil M and PekaÅ™ M. Effect of new hydrophobic modification of hyaluronan on its solution properties: evaluation of self-aggregation. Polym.2009, 76(3): 443-448.
- Vinogradov A.M., Winston M., Rupp C.J., Stoodley P. Rheology of biofilms formed from the dental plaque pathogen Streptococcus mutans. Biofilms. 2004.1: 49–56
- Iannitti, T., Bingöl, A.Ö., Rottigni, V., Palmieri, B., A new highly viscoelastic hyaluronic acid gel: Rheological properties, biocompatibility and clinical investigation in esthetic and restorative surgery. Int. J. Pharm. 2013.456, 583–592.
- Li and Mooney, 2016 Designing hydrogels for controlled drug delivery. Nat Rev Mater. 2016; 1(12): 16071.
- Scalzo, M.; Orlandi, C.; Simonetti, N.; Cerreto, F. Study of interaction effects of polyacrylic acid polymers (carbopol 940) on antimicrobial activity of methyl parahydroxybenzoate against some gram-negative, gram-positive bacteria and yeast. J Pharm Pharmacol. 1996, 48(11), 1201-1205. doi: 10.1111/j.2042-7158.1996. tb 03921.x.
- Wang YY, Hong CT, Chiu WT, Fang JY. In vitro and in vivo evaluations of topically applied capsaicin and nonivamide from hydrogels. International journal of pharmaceutics. 2001; 224:89–104.
Reviewer 2:
- The study of this new formulation is serious and rigorous but it could only be accepted for publication if the improvement of this hydrogel with respect to other already published formulations is highlighted.
Response to Reviewer 2: The suggested corrections have been carried in the manuscript as follows:
- Conclusions
This paper presents the results of a good form of hydrogel formulation of capsaicin extract. The findings about the inner structure of the gel conclude for a pseudo-plastic form with good rheological properties.
The in vitro evaluation of drug release suggests a prolong release and good bioavailability of this formulation and may have great relevance in the local treatment of inflammatory joints diseases. Wang et al., (2001) have evaluated the skin absorption of capsaicin from different hydrogels and we may conclude that the in vitro permeation of capsaicin from hydrogels depends on the physicochemical nature and the concentration of the polymer used. They emphasize that higher levels of capsaicin are released from anionic polymer hydrogels than cream bases [45].
From the antimicrobial and antifungal properties assessment we may concluded that our formulation has good antimicrobial effects against Escherichia coli and same effects between Bacillus cereus and Salmonella typhi compare to pure capsaicin. Also, the Carbopol formulation exhibits good antifungal properties from each species of Candida especially from Candida albicans.
According these results we strongly recommend this formulation for dermatological uses not only for anti-inflammatory properties by also for antimicrobial ones.

Round 2
Reviewer 2 Report
I believe that the manuscript can be accepted in the present format